# Selective Mixup Helps with Distribution Shifts, But Not (Only) because of Mixup

## Abstract

**Context.** Mixup is a highly successful technique to improve generalization of neural networks by augmenting the training data with combinations of random pairs. Selective mixup is a family of methods that apply mixup to specific pairs, e.g. only combining examples across classes or domains. These methods have claimed remarkable improvements on benchmarks with distribution shifts, but their mechanisms and limitations remain poorly understood.

**Findings.** We examine an overlooked aspect of selective mixup that explains its success in a completely new light. We find that the non-random selection of pairs affects the training distribution and improve generalization by means completely unrelated to the mixing. For example in binary classification, mixup across classes implicitly resamples the data for a uniform class distribution — a classical solution to label shift. We show empirically that this implicit resampling explains much of the improvements in prior work. Theoretically, these results rely on a "regression toward the mean", an accidental property that we identify in several datasets.

**Takeaways.** We have found a new equivalence between two successful methods: selective mixup and resampling. We identify limits of the former, confirm the effectiveness of the latter, and find better combinations of their respective benefits.

## 1 Introduction

Mixup and its variants are some of the few methods that improve generalization across tasks and modalities with no domain-specific information [36]. Standard mixup replaces training data with linear combinations of random pairs of examples, proving successful e.g. for image classification [35], semantic segmentation [9], natural language processing [30], and speech processing [21].

This paper focuses on scenarios of distribution shift and on variants of mixup that improve out-of-distribution (OOD) generalization. We examine the family of methods that apply mixup on selected pairs of examples, which we refer to as *selective mixup* [7, 15, 19, 22, 28, 31, 33]. Each of these method uses a predefined criterion.[1] For example, some methods combine examples across classes [33] (Figure 1) or across domains [31, 15, 19]. These simple heuristics have claimed remarkable improvements on benchmarks such as DomainBed [5], WILDS [12], and Wild-Time [32].

Despite impressive empirical performance, the theoretical mechanisms of selective mixup remain obscure. For example, the selection criteria proposed in [33] include the selection of pairs of the same class / different domains, but also the exact opposite.

---

[1]We focus on the basic implementation of selective mixup as described by Yao et al. [33], i.e. without additional regularizers or modifications of the learning objective described in various other papers.

Submitted to 37th Conference on Neural Information Processing Systems (NeurIPS 2023). Do not distribute.

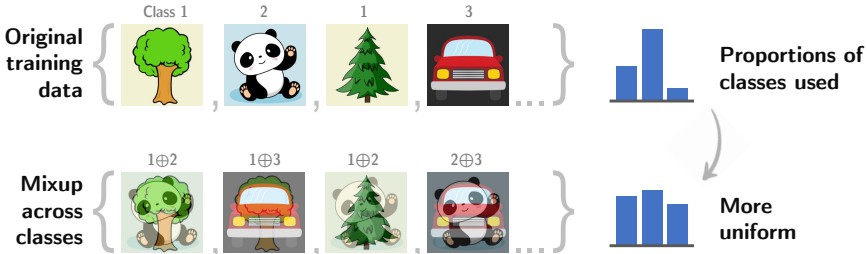

Figure 1: Selective mixup is a family of methods that replace the training data with combined pairs of examples fulfilling a predefined criterion, e.g. pairs of different classes. As an overlooked side effect, this modifies the training distribution: in this case, sampling classes more uniformly. This effect is responsible for much of the resulting improvements in OOD generalization.

This raises questions:

1. What mechanisms are responsible for the improvements of selective mixup?

2. What makes each selection criterion suitable to any specific dataset?

This paper presents surprising answers by highlighting an overlooked side effect of selective mixup. **The non-random selection of pairs implicitly biases the training distribution and improve generalization by means completely unrelated to the mixing**. We observe empirically that simply concatenating – rather than mixing – the selected pairs along the mini-batch dimension often produces the same improvements as mixing them. This critical ablation was absent from prior studies.

We also analyze theoretically the resampling induced by different selection criteria. We find that conditioning on a "different attribute" (e.g. combining examples across classes or domains) brings the training distribution of this attribute closer to a uniform one. Consequently, the imbalances in the data often "regress toward the mean" with selective mixup. We verify empirically that several datasets do indeed shift toward a uniform class distribution in their test split (see Figure 10). We also find remarkable correlation between improvements in performance and the reduction in divergence of training/test distributions due to selective mixup. This also predicts an unknown failure mode of selective mixup when the above property does not hold.

**Our contributions are summarized as follows.**

- We point out an overlooked resampling effect when applying selective mixup (Section 3).

- We show theoretically that certain selection criteria induce a bias in the distribution of features and/or classes equivalent to a "regression toward the mean" (Theorem 3.1). In binary classification for example, selecting pairs across classes is equivalent to sampling uniformly over classes, the standard approach to address label shift and imbalanced data.

- We verify empirically that multiple datasets indeed contain a regression toward a uniform class distribution across training and test splits (Section 4.6). We also find that improvements from selective mixup correlate with reductions in divergence of training/test distributions over labels and/or covariates. This strongly suggests that resampling is the main driver for these improvements.

- We compare many selection criteria and resampling baselines on five datasets. In all cases, improvements with selective mixup are partly or fully explained by resampling effects (Section 4).

**The implications for future research are summarized as follows.**

- We connect two areas of the literature by showing that selective mixup is sometimes equivalent to resampling, a classical strategy for distribution shifts [3, 8]. This hints at possible benefits from advanced methods for label shift and domain adaptation on benchmarks with distribution shifts.

- The resampling explains why different criteria in selective mixup benefit different datasets: they affect the distribution of features and/or labels and therefore address covariate and/or label shift.

- There is a risk of overfitting to the benchmarks: we show that much of the observed improvements rely on the accidental property of a "regression toward the mean" in the datasets examined.

## 2 Background: mixup and selective mixup

**Notations.** We consider a classification model $f_{\boldsymbol{\theta}} : \mathbb{R}^d \to [0, 1]^C$ of learned parameters $\boldsymbol{\theta}$. It maps an input vector $\boldsymbol{x} \in \mathbb{R}^d$ to a vector $\boldsymbol{y}$ of scores over $C$ classes. The training data for such a model is typically a set of labeled examples $\mathcal{D} = \{(\boldsymbol{x}_i, \boldsymbol{y}_i, d_i)\}_{i=1}^n$ where $\boldsymbol{y}_i$ are one-hot vectors encoding ground-truth labels, and $d_i \in \mathbb{N}$ are optional discrete domain indices. Domain labels are sometimes available e.g. in datasets with different image styles [14] or collected over different time periods [12].

**Training with ERM.** Standard empirical risk minimization (ERM) optimizes the model's parameters for $\min_{\boldsymbol{\theta}} \mathcal{R}(f_{\boldsymbol{\theta}}, \mathcal{D})$ where the expected training risk, for a chosen loss function $\mathcal{L}$, is defined as:

$$\mathcal{R}(f_{\boldsymbol{\theta}}, \mathcal{D}) = \mathbb{E}_{(\boldsymbol{x}, \boldsymbol{y}) \in \mathcal{D}} \, \mathcal{L}\big(f_{\boldsymbol{\theta}}(\boldsymbol{x}), \boldsymbol{y}\big). \tag{1}$$

An empirical estimate is obtained with an arithmetic mean over instances of the dataset $\mathcal{D}$.

**Training with mixup.** Standard mixup essentially replaces training examples with linear combinations of random pairs in both input and label space. We formalize it by redefining the training risk.

$$\mathcal{R}_{\mathrm{mixup}}(f_{\boldsymbol{\theta}}, \mathcal{D}) = \mathbb{E}_{(\boldsymbol{x}, \boldsymbol{y}) \in \mathcal{D}} \, \mathcal{L}\big(f(c\,\boldsymbol{x} + (1-c)\widetilde{\boldsymbol{x}}, \ c\,\boldsymbol{y} + (1-c)\,\widetilde{\boldsymbol{y}})\big) \tag{2}$$

$$\text{with mixing coefficients } c \sim \mathcal{B}(2, 2) \quad \text{and paired examples } (\widetilde{\boldsymbol{x}}, \widetilde{\boldsymbol{y}}) \sim \mathcal{D}. \tag{3}$$

The expectation is approximated by sampling different coefficients and pairs at every training iteration.

**Selective mixup.** While standard mixup combines random pairs, selective mixup only combines pairs that fulfill a predefined criterion. To select these pairs, the method starts with the original data $\mathcal{D}$, then for every $(\boldsymbol{x}, \boldsymbol{y}, d) \in \mathcal{D}$ it selects a $(\widetilde{\boldsymbol{x}}, \widetilde{\boldsymbol{y}}, \widetilde{d}) \in \mathcal{D}$ such that they fulfill the criterion represented by the predicate $\mathrm{Paired}(\cdot, \cdot)$. For example, the criterion *same class / different domain* a.k.a. "intra-label LISA" in [33] is implemented as follows:

$$\mathrm{Paired}\big((\boldsymbol{x}_i, \boldsymbol{y}_i, d_i), (\widetilde{\boldsymbol{x}}_i, \widetilde{\boldsymbol{y}}_i, \widetilde{d}_i)\big) = true \ \text{ iff } \ (\widetilde{\boldsymbol{y}} = \boldsymbol{y}) \wedge (\widetilde{d} \neq d) \ \textit{(same class / diff. domain)} \tag{4a}$$

Other examples:

$$\mathrm{Paired}\big((\boldsymbol{x}_i, \boldsymbol{y}_i, d_i), (\widetilde{\boldsymbol{x}}_i, \widetilde{\boldsymbol{y}}_i, \widetilde{d}_i)\big) = true \ \text{ iff } \ (\widetilde{\boldsymbol{y}} \neq \boldsymbol{y}) \qquad\qquad \textit{(different class)} \tag{4b}$$

$$\mathrm{Paired}\big((\boldsymbol{x}_i, \boldsymbol{y}_i, d_i), (\widetilde{\boldsymbol{x}}_i, \widetilde{\boldsymbol{y}}_i, \widetilde{d}_i)\big) = true \ \text{ iff } \ (\widetilde{d} = d) \qquad\qquad \textit{(same domain)} \tag{4c}$$

## 3 Selective mixup modifies the training distribution

The new claims of this paper comprise two parts.

1. Estimating the training risk with selective mixup (Eq. 2) uses a different sampling of examples from $\mathcal{D}$ than ERM (Eq. 1). We demonstrate it theoretically in this section.

2. We hypothesize that the biased sampling of training examples influences the generalization properties of the learned model, regardless of the mixing operation. We verify this empirically in Section 4 using ablations of selective mixup that omit the mixing operation — a critical baseline absent from previous studies.

**Training distribution.** This distribution refers to the examples sampled from $\mathcal{D}$ to estimate the training risk (Eq. 1 or 2) — whether these are then mixed or not. The following discussion focuses on distributions over classes ($\boldsymbol{y}$) but analogous arguments apply to covariates ($\boldsymbol{x}$) and domains ($d$).

**With ERM**, the training distribution equals the dataset distribution because the expectation in Eq. (1) is over uniform samples of $\mathcal{D}$. We obtain an empirical estimate by averaging all one-hot labels, giving the vector of discrete probabilities $\mathbf{p}_{\mathrm{Y}}(\mathcal{D}) = \oplus_{(\boldsymbol{x}, \boldsymbol{y}) \in \mathcal{D}} \, \boldsymbol{y} \, / \, |\mathcal{D}|$ where $\oplus$ is the element-wise sum.

**With selective mixup**, evaluating the risk (Eq. 2) requires pairs of samples. The first element of a pair is sampled uniformly, yielding the same $\mathbf{p}_{\mathrm{Y}}(\mathcal{D})$ as ERM. The second element is selected as described above, using the first element and one chosen predicate $\mathrm{Paired}(\cdot, \cdot)$ e.g. from (4a–4c). For our analysis, we denote these "second elements" of the pairs as the virtual data:

$$\widetilde{\mathcal{D}} = \big\{ (\widetilde{\boldsymbol{x}}_i, \widetilde{\boldsymbol{y}}_i, \widetilde{d}_i) \sim \mathcal{D} : \ \mathrm{Paired}\big((\boldsymbol{x}_i, \boldsymbol{y}_i, d_i), (\widetilde{\boldsymbol{x}}_i, \widetilde{\boldsymbol{y}}_i, \widetilde{d}_i)\big) = true, \ \ \forall \, i = 1, \dots, |\mathcal{D}| \big\}. \tag{5}$$

We can now analyze the overall training distribution of selective mixup. An empirical estimate is obtained by combining the distributions resulting from the two elements of the pairs, which gives the vector $\mathbf{p}_{\mathrm{Y}}(\mathcal{D} \cup \widetilde{\mathcal{D}}) = \big(\mathbf{p}_{\mathrm{Y}}(\mathcal{D}) \oplus \mathbf{p}_{\mathrm{Y}}(\widetilde{\mathcal{D}})\big) / 2$.

**Regression toward the mean.** With the criterion *same class*, it is obvious that $\mathbf{p}_Y(\widetilde{\mathcal{D}}) = \mathbf{p}_Y(\mathcal{D})$. Therefore these variants of selective mixup are not concerned with resampling effects.[2] In contrast, the criteria *different class* or *different domain* do bias the sampling. In the case of binary classification, we have $\mathbf{p}_Y(\widetilde{\mathcal{D}}) = 1 - \mathbf{p}_Y(\mathcal{D})$ and therefore $\mathbf{p}_Y(\mathcal{D} \cup \widetilde{\mathcal{D}})$ is uniform. This means that selective mixup with the *different class* criterion has the side effect of balancing the training distribution of classes, a classical mitigation of class imbalance [10, 13]. For multiple classes, we have a more general result.

**Theorem 3.1.** *Given a dataset $\mathcal{D} = \{(\boldsymbol{x}_i, \boldsymbol{y}_i)\}_i$ and paired data $\widetilde{\mathcal{D}}$ sampled according to the "different class" criterion, i.e. $\widetilde{\mathcal{D}} = \{(\widetilde{\boldsymbol{x}}_i, \widetilde{\boldsymbol{y}}_i) \sim \mathcal{D} \ \text{s.t.} \ \widetilde{\boldsymbol{y}}_i \neq \boldsymbol{y}_i\}$, then the distribution of classes in $\mathcal{D} \cup \widetilde{\mathcal{D}}$ is more uniform than in $\mathcal{D}$. Formally, the entropy $\mathbb{H}(\mathbf{p}_Y(\mathcal{D})) \leq \mathbb{H}(\mathbf{p}_Y(\mathcal{D} \cup \widetilde{\mathcal{D}}))$.*

*Proof:* see Appendix C.

Theorem 3.1 readily extends in two ways. First, the same effect also results from the *different domain* criterion: if each domain contains a different class distribution, the resampling from this criterion averages them out, yielding a more uniform aggregated training distribution. Second, this averaging applies not only to class labels ($\boldsymbol{y}$) but also covariates ($\boldsymbol{x}$). An analysis using distributions is ill-suited but the mechanism similarly affects the sampling of covariates when training with selective mixup.

**When does one benefit from the resampling (regardless of mixup)?** The above results mean that selective mixup can implicitly reduce imbalances (a.k.a. biases) in the training data. When these are not spurious and also exist in the test data, the effect on predictive performance could be detrimental.

We expect benefits (which we verify in Section 4) on datasets with distribution shifts, whose training/test splits contain different imbalances by definition. Softening imbalances in the training data is then likely to bring the training and test distributions closer to one another, in particular with extreme shifts such as the complete reversal of a spurious correlation (e.g. *waterbirds* [24], Section 4.1).

We also expect a benefit on worst-group metrics (e.g. with *civilComments* [12] in Section 4.4). The challenge in these datasets comes from the imbalance of class/domain combinations in the training data, and previous work has indeed shown that balancing is a competitive baseline [8, 24].

# 4  Experiments

We performed a large number of experiments to understand the contribution of the different effects of selective mixup and other resampling baselines (see Appendix B for complete results).

**Datasets.** We focus on five datasets that previously showed improvements with selective mixup. We selected them to cover a range of modalities (vision, NLP, tabular), settings (binary, multiclass), and types of distribution shifts (covariate, label, and subpopulation shifts).

- **Waterbirds** [24] is a popular artificial dataset used to study distribution shifts. The task is to classify images of birds into two types. The image backgrounds are also of two types, and the correlation between birds and backgrounds is reversed across the training and test splits. The type of background in each image serves as its domain label.

- **CivilComments** [12] is a widely-used dataset of online text comments to be classified as toxic or not. Each example is labeled with a topical attribute (e.g. Christian, male, LGBT, etc.) that is spuriously associated with ground truth labels in the training data. These attributes serve as domain labels. The target metric is the worst-group accuracy where the groups correspond to all toxicity/attribute combinations.

- **Wild-Time Yearbook** [32] contains yearbook portraits to be classified as male or female. It is part of the Wild-Time benchmark, which is a collection of real-world datasets captured over time. Each example belongs to a discrete time period that serves as its domain label. Distinct time periods are assigned to the training and OOD test splits (see Figure 10).

- **Wild-Time arXiv** [32] contains titles of arXiv preprints. The task is to predict each paper's primary category among 172 classes. Time periods serve as domain labels.

- **Wild-Time MIMIC-Readmission** [32] contains hospital records (sequences of codes representing diagnoses and treatments) to be classified into two classes. The positive class indicates the readmission of the patient at the hospital within 15 days. Time periods serve as domain labels.

---

[2]The absence of resampling effects holds for *same class* and *same domain* alone, but not in conjunction with other criteria. See e.g. the differences between *same domain / diff. class* and *any domain / diff. class* in Figure 3.

157 **Methods.** We train standard architectures suited to each dataset with the methods below (details
158 in Appendix A). We perform early stopping i.e. recording metrics for each run at the epoch of
159 highest ID or worst-group validation performance (for *Wild-Time* and *waterbirds*/*civilComments*
160 datasets respectively). We plot average metrics in bar charts over 9 different seeds with error bars
161 representing $\pm$ one standard deviation. **ERM** and **vanilla mixup** are the standard baselines. Baseline
162 **resampling** uses training examples with equal probability from each class, domain, or combinations
163 thereof as in [8, 24]. **Selective mixup** (■) includes all possible selection criteria based on classes
164 and domains. We avoid ambiguous terminology from earlier works because of inconsistent usage
165 (e.g. "intra-label LISA" means "different domain" in [12] but not in [32]). **Selective sampling** (■) is
166 a novel ablation of selective mixup where the selected pairs are not mixed, but concatenated along
167 the mini-batch dimension. Half of the pairs are dropped at random to keep the size of mini-batches
168 constant. Therefore any difference between selective sampling and ERM is attributable only to
169 resampling effects. We also include **novel combinations** (■) of sampling and mixup. Code to
170 reproduce our experiments and figures: `https://github.com/<anonymized>/<anonymized>`.

## 4.1 Results on the *waterbirds* dataset

172 The target metric for this dataset is the worst-group accuracy, with groups defined as the four
173 class/domain combinations. The two difficulties are (1) a class imbalance (77/23%) and (2) a
174 correlation shift (spurious class/domain association reversed at test time). See discussion in Figure 2.

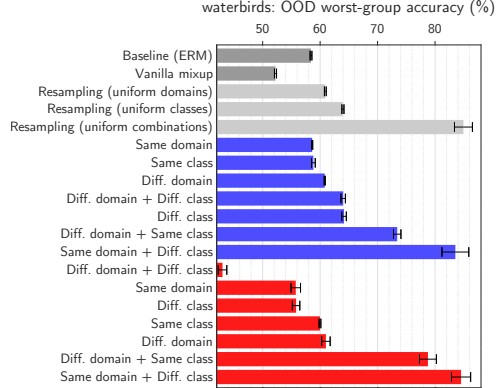

Figure 2: Main results on *waterbirds*.

We first observe that vanilla mixup is detrimental compared to ERM. Resampling with uniform class/domain combinations is hugely beneficial, for the reasons explained in Figure 3. The ranking of various criteria for selective sampling is similar whether with or without mixup. Most interestingly, the best criterion performs similarly, but no better than the best resampling.

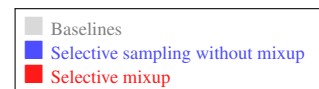

175 This suggest that **the excellent performance of the best version of selective mixup is here entirely**
176 **due to resampling**. Note that the efficacy of resampling on this dataset is not a new finding [8, 24].
177 What is new is its equivalence with the best variant of selective mixup. We further explain this claim
178 in Figure 3 by examining the proportions of classes and domains sampled by each training method.

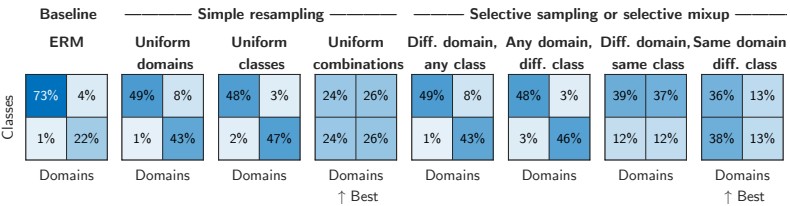

**Resampling uniform combinations** gives them all equal weights, just like the worst-group target metric.
**Selective mixup with same domain/diff. class** also gives equal weights to the classes, while breaking
the spurious pattern between groups and classes, unlike any other criterion.

Figure 3: The sampling ratios of each class/domain clearly explain the performance of the best methods (*waterbirds*).

## 4.2 Results on the *yearbook* dataset

180 The difficulty of this dataset comes from a slight class imbalance and the presence of covariate/label
181 shift (see Figure 10). The test split contains several domains (time periods). The target metric is the
182 worst-domain accuracy. Figure 4 shows that vanilla mixup is slightly detrimental compared to ERM.
183 Resampling for uniform classes gives a clear improvement because of the class imbalance. With
184 selective sampling (no mixup), the only criteria that improve over ERM contain "different class".
185 This is expected because this criterion implicitly resamples for a uniform class distribution.

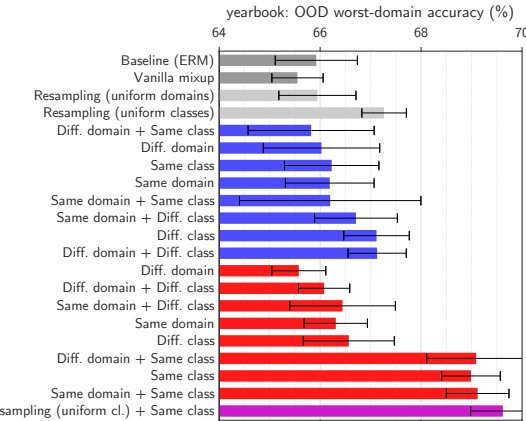

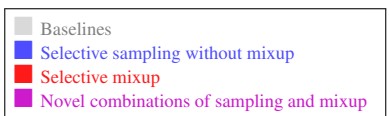

Figure 4: Main results on *yearbook*.

With selective mixup, the "different class" criterion is not useful, but "same class" performs significantly better than ERM. Since this criterion alone does not have resampling effects, it indicates a genuine benefit from mixup restricted to pairs of the same class.

To investigate whether some of the improvements are due to resampling, we measure the divergence between training and test distributions of classes and covariates (details in Appendix A). Figure 5) shows first that there is a clear variation among different criteria (● blue dots) i.e. some bring the training/test distributions closer to one another. Second, there is a remarkable correlation between the test accuracy and the divergence, on both classes and covariates.[3] This means that resampling effects do occur and also play a part in the best variants of selective mixup.

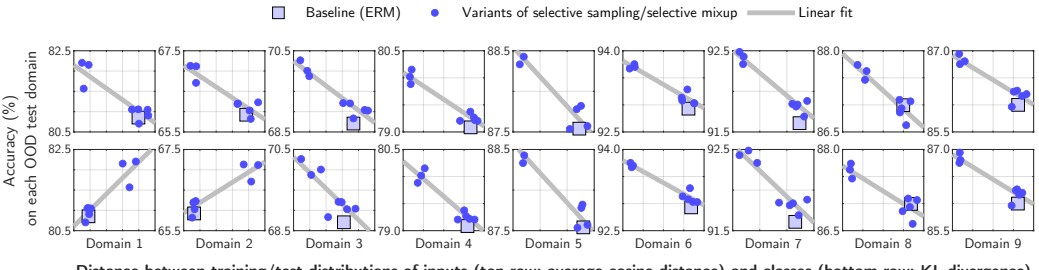

Figure 5: Different selection criteria (●) modify the distribution of both covariates and labels (upper and lower rows). The resulting reductions in divergence between training and test distributions correlate remarkably well with test performance.[3] This confirms the contribution of resampling to the overall performance of selective mixup.

Finally, the improvements from simple resampling and the best variant of selective mixup suggest a new combination. We train a model with uniform class sampling and selective mixup using the "same class" criterion, and obtain performance superior to all existing results (last row in Figure 5). This confirms the **complementarity of the effects of resampling and within-class selective mixup**.

## 4.3 Results on the *arXiv* dataset

This dataset has difficulties similar to *yearbook* and also many more classes (172). Simple resampling for uniform classes is very bad (literally off the chart in Figure 6) because it overcorrects the imbalance (the test distribution being closer to the training than to a uniform one). Uniform *domains* is much better since its effect is similar but milder.

All variants of selective mixup (🟥) perform very well, but they improve over ERM even without mixup (🟦). And the selection criteria rank similarly with or without mixup, suggesting that parts of the improvements of selective mixup is due to the resampling. Given that vanilla mixup also clearly improves over ERM, the performance of **selective mixup is explained by cumulative effects of vanilla mixup and resampling effects**. This also suggests new combinations of methods (🟪) among which we find one version marginally better than the best variant of selective mixup (last row).

---

[3]As expected, the correlation is reversed for the first two test domains in Figure 5 since they are even further from a uniform class distribution than the average of the training data, as seen in Figure 10.

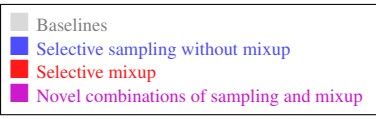

Figure 6: Main results on *arXiv*.

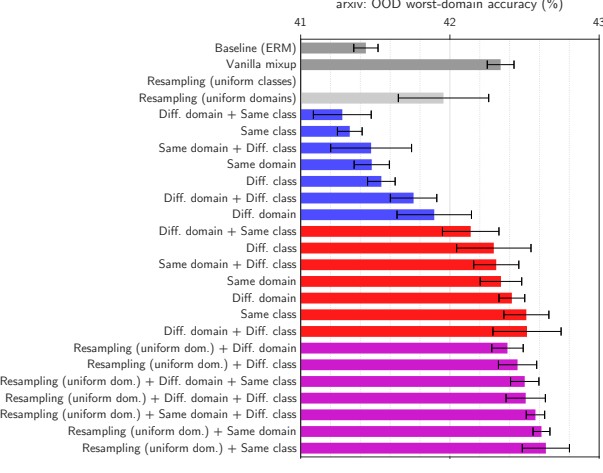

To investigate the contribution of resampling, we measure the divergence between training/test class distributions and plot them against the test accuracy (Figure 7). We observe a strong correlation across methods. Mixup essentially offsets the performance by a constant factor. This suggests again the independence of the effects of mixup and resampling.

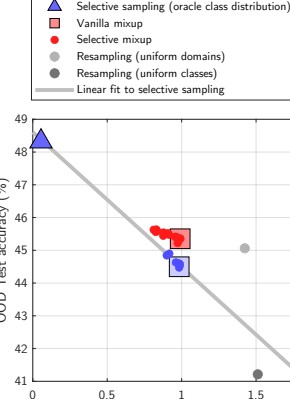

Figure 7: Divergence of tr./test class distributions vs. test accuracy.

The resampling baselines (●●) also roughly agree with a linear fit to the "selective sampling" points. We therefore hypothesize that **all these methods are mostly addressing label shift**. We verify this hypothesis with the remarkable fit of an additional point (▲) of a model trained by resampling according to the test set class distribution, i.e. cheating.

It represents an upper bound that might be achievable in future work with methods for label shift [1, 17].

We replicated these observations on every test domain of this dataset (Figure 14 in the appendix).

## 4.4 Results on the *civilComments* dataset

This dataset mimics a subpopulation shift because the worst-group metric requires high accuracy on classes and domains under-represented in the training data. It also contains an implicit correlation shift because any class/domain association (e.g. "Christian" comments labeled as toxic more often than not) becomes spurious when evaluating individual class/domain combinations.

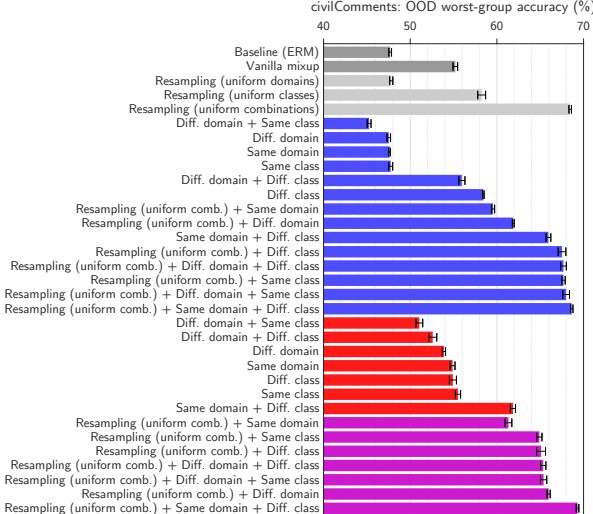

Figure 8: Main results on *civilComments*.

For the above reasons, it makes sense that resampling for uniform classes or combinations greatly improves performance, as shown in prior work [8].

With selective mixup (🟥), some criterion (same domain/diff. class) performs clearly above all others. But it works **even better without mixup**! (🟦) Among many other variations, **none surpasses the uniform-combinations baseline**.

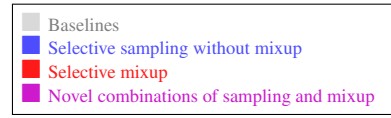

## 4.5 Results on the *MIMIC-Readmission* dataset

This dataset contains a class imbalance (about 78/22% in training data), label shift (the distribution being more balanced in the test split), and possibly covariate shift. It is unclear whether the task is causal or anticausal (labels causing the features) because the inputs contain both diagnoses and treatments. The target metric is the area under the ROC curve (AUROC) which gives equal importance to both classes. We report the worst-domain AUROC, i.e. the lowest value across test time periods.

Vanilla mixup performs a bit better than ERM. Because of the class imbalance, resampling for uniform classes also improves ERM. As expected, this is perfectly equivalent to the selective sampling criterion "diffClass" and they perform therefore equally well. Adding mixup is yet a bit better, which suggests again that **the performance of selective mixup is merely the result of the independent effects of vanilla mixup and resampling**. We further verify this explanation with the novel combination of simple resampling and vanilla mixup, and observe almost no difference whether the mixing operation is performed or not (last two rows in Figure 9).

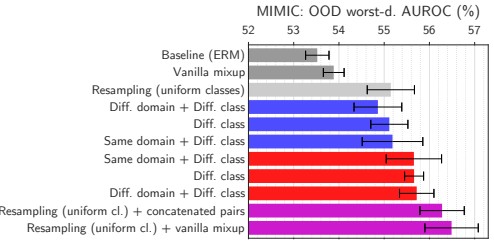

Figure 9: Main results on *MIMIC-Readmission*.

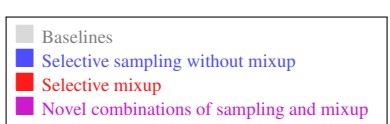

To further support the claim that these methods mostly address label shift, we report in Table 1 the proportion of the majority class in the training and test data. We observe that the distribution sampled by the best training methods brings it much closer to that of the test data.

| Proportion of majority class | (%) |
|---|---|
| In the dataset (training) | 78.2 |
| In the dataset (validation) | 77.8 |
| In the dataset (OOD test) | 66.5 |
| **Sampled by different training methods** | |
| Resampling (uniform classes) | 50.0 |
| Diff. domain + diff. class | 50.0 |
| Diff. class | 50.1 |
| Same domain + Diff. class | 49.9 |
| Resampling (uniform cl.) + concatenated pairs | 64.3 |
| Resampling (uniform cl.) + vanilla mixup | 64.3 |

Table 1: The performance of the various methods on *MIMIC-Readmission* is explained by their correction of a class imbalance. The best training methods (boxed numbers) sample the majority class in a proportion much closer to that of the test data.

## 4.6 Evidence of a "regression toward the mean" in the data

We hypothesized in Section 3 that the resampling benefits are due to a "regression toward the mean" across training and test splits. We now check for this property and find indeed a shift toward uniform class distributions in all datasets studied. For the Wild-Time datasets, we plot in Figure 10 the ratio of the minority class (binary tasks: yearbook, MIMIC) and class distribution entropy (multiclass task: arxiv). Finding this property in all three datasets agrees with the proposed explanation and the fact that we selected them because they previously showed improvements with selective mixup in [32].

In the *waterbirds* and *civilComments* datasets, the shift toward uniformity¨also holds, but artificially. The training data contains imbalanced groups (class/domain combinations) whereas the evaluation with worst-group accuracy implicitly gives uniform importance to all groups.

# 5 Related work

**Mixup and variants.** Mixup was originally introduced in [36] and numerous variants followed [2]. Many propose modality-specific mixing operations: CutMix [34] replaces linear combinations with collages of image patches, Fmix [6] combines image regions based on frequency contents, AlignMixup [29] combines images after spatial alignment. Manifold-mixup [30] replaces the mixing in input space with the mixing of learned representations, making it applicable to text embeddings.

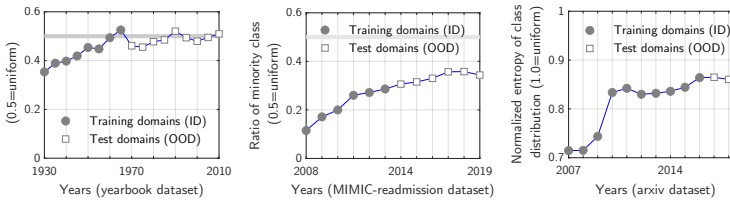

Figure 10: The class distribution shifts toward uniformity in these Wild-Time datasets, which agrees with the explanation that resampling benefits are due to a "regression toward the mean".

**Mixup for OOD generalization.** Mixup has been integrated into existing techniques for domain adaptation (DomainMix [31]), domain generalization (FIXED [20]), and with meta learning (Reg-Mixup [23]). This paper focuses on variants we call "*selective mixup*" that use non-uniform sampling of the pairs of mixed examples. LISA [33] proposes two heuristics, same-class/different-domain and vice versa, used in proportions tuned by cross-validation on each dataset. Palakkadavath et al. [22] use same-class pairs and an additional objective to encourage invariance of the representations to the mixing. CIFair [28] uses same-class pairs with a contrastive objective to improve algorithmic fairness. SelecMix [7] proposes a selection heuristic to handle biased training data: same class/different biased attribute, or vice versa. DomainMix [31] uses different-domain pairs for domain adaptation. DRE [15] uses same-class/different-domain pairs and regularize their Grad-CAM explanations to improve OOD generalization. SDMix [19] applies mixup on examples from different domains with other improvements to improve cross-domain generalization for activity recognition.

**Explaining the benefits of mixup** has invoked regularization [37] and augmentation [11] effects, the introduction of label noise [18], and the learning of rare features [38]. These works focus on the mixing and in-domain generalization, whereas we focus on the selection and OOD generalization.

**Training on resampled data.** We find that selective mixup is sometimes equivalent to training on resampled or reweighted data. Both are standard tools [10, 13] to handle distribution shifts in a domain adaptation setting, also known as importance-weighted empirical risk minimization (IW-ERM) [25, 4]. For covariate shift, IW-ERM trains a model with a weight or sampling probability on each training point $x$ as its likelihood ratio $p_{\text{target}}(x)/p_{\text{source}}(x)$. Likewise with labels $y$ and $p_{\text{target}}(y)/p_{\text{source}}(y)$ for label shift [1, 17], Recently, [8, 24] showed that reweighting and resampling are competitive with the state of the art on multiple OOD and label-shift benchmarks [3].

## 6   Conclusions and open questions

**Conclusions.** This paper helps understand selective mixup, which is one of the most successful and general methods for distribution shifts. We showed unambiguously that much of the improvements were actually unrelated to the mixing operation and could be obtained with much simpler, well-known resampling methods. On datasets where mixup does bring benefits, we could then obtain even better results by combining the independent effects of the best mixup and resampling variants.

**Limitations.** We focused on the simplest version selective mixup as described by Yao et al. [33]. Many papers combine the principle with modifications to the learning objective [7, 15, 19, 22, 28, 31]. Resampling likely plays a role in these methods too but this claim requires further investigation.

We evaluated "only" five datasets. Since we introduced simple ablations that can single out the effects of resampling, we hope to see future re-evaluations of other datasets.

Because we selected datasets that had previously shown benefits with selective mixup, we could not verify the predicted failure mode when there is no "regression toward the mean" in the data.

Finally, this work is **not** about designing new algorithms to surpass the state of the art. Our focus is on improving the scientific understanding of existing mixup strategies and their limitations.

**Open questions.** Our results leave open the question of the applicability of selective mixup to real situations. The "regression toward the mean" explanation indicates that much of the observed improvements are accidental since they rely on an artefact of some datasets. In real deployments, distribution shifts cannot be foreseen in nature nor magnitude. This is a reminder of the relevance of Goodhart's law to machine learning [26] and of the risk of overfitting to popular benchmarks [16].

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
