# Selective Mixup Helps with Distribution Shifts, But Not (Only) because of Mixup

# Appendices

## A  Experimental details

We follow prior work on each dataset for the **architectures and hyperparameters** of our experiments. For each dataset, all methods compared use hyperparameters initially validated with the ERM baseline. All experiments use early stopping i.e. recording metrics for each run at the epoch of highest ID or worst-group validation performance (for *Wild-Time* and *waterbirds/civilComments* datasets respectively). Each dataset/method is run with 9 different seeds unless otherwise noted. The bar charts report the average over these seeds and error bars represent $\pm$ one standard deviation.

We noticed that there is sometimes considerable **variability in the results reported in prior work** even for datasets/methods supposedly identical (e.g. resampling baselines on *waterbirds*). Therefore we only make comparisons across results obtained within a unique code base after re-running all baselines in a comparable setting. Exact hyperparameters for all experiments can be found in our code: https://github.com/<anonymized>/<anonymized>.

We also found some **issues in existing code** that we could not clear up with their authors despite multiple requests. This includes inconsistent preprocessing and duplicated data in the preprocessing of civilComments in [8], "magic constants" in the implementation of selective mixup (LISA) in [33], inappropriate architectures for *MIMIC* in [32]. We fixed these issues in our codebase. Therefore we refrain from claims or direct comparisons with the absolute state of the art.

Dataset-specific notes:

- On *waterbirds*, we use ImageNet-pretrained ResNet-50 models. The results in the main paper use linear classifiers trained on frozen features. We report similar results with fine-tuned ResNet-50 models in Figure 11.

- On *CivilComments*, we use a standard pretrained BERT. To limit the computational expense for our large number of experiments, we use the BERT-tiny version (2 layers, 2 attention heads, embeddings of size 128). The results in the main paper use linear classifiers on frozen features. We report similar results with fine-tuned models in Figure 16 (using only one seed).

- On *Wild-Time Yearbook*, we train the small CNN architecture described in [32] from scratch. In the analysis of Figure 5, we measure the distance between the training and test distributions of inputs (vectorized grayscale images). To do so, we measure the distance between every pair across the two sets. For each test example, we keep the minimum distance (i.e. closest training example), then average these distances over the test set.

- On *Wild-Time arXiv*, we use random subset of 10% of the dataset. We verified on a small number of experiments that this produces very similar results to the full dataset at a fraction of the computational expense.

- On *Wild-Time MIMIC-Readmission*, the baseline transformer architecture proposed in [33] seems inappropriate. Its ID and OOD performance is surpassed by random guessing or even by constant predictions of the majority training class. The issue probably went unnoticed because the standard accuracy metric is misleading with imbalanced data (70% ID accuracy of that ERM baseline is worse than chance).
  To remedy this, we first switch to the AUROC metric. It gives equal weight to the classes and 50% is then unambiguously equivalent to random chance.
  Second, we use a much simpler architecture. We train a "bag of embeddings" where each token (diagnosis/treatment code) is assigned a learned embedding, which are summed across sequences then fed to a linear classifier.

All experiments were run on a single laptop with an Nvidia GeForce RTX 3050 Ti GPU.

# B    Additional results

We show below results from the main paper while including in-domain (ID), out-of-distribution (OOD) average-domain/average-group, and OOD worst-domain/worst-group performance. The OOD metrics are always strongly correlated across methods and training epochs, but ID and OOD performance sometimes require a trade-off, as noted recently in [27].

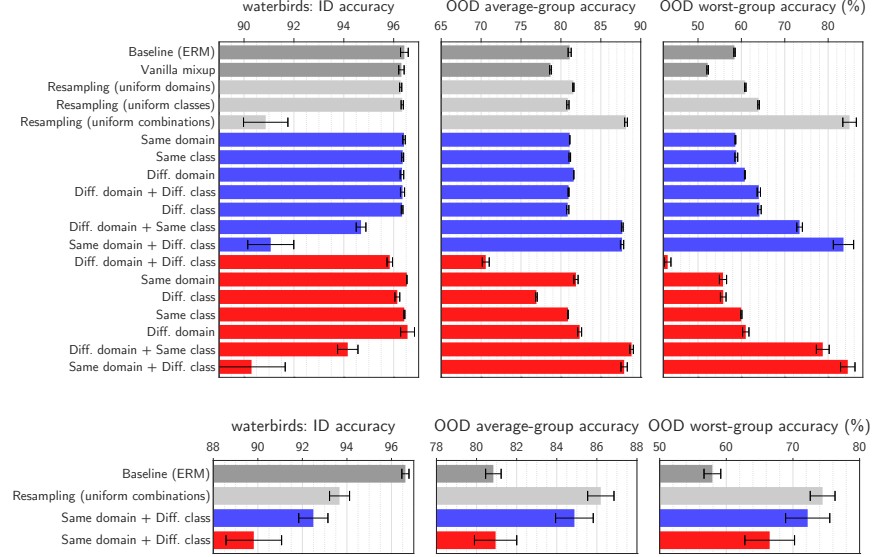

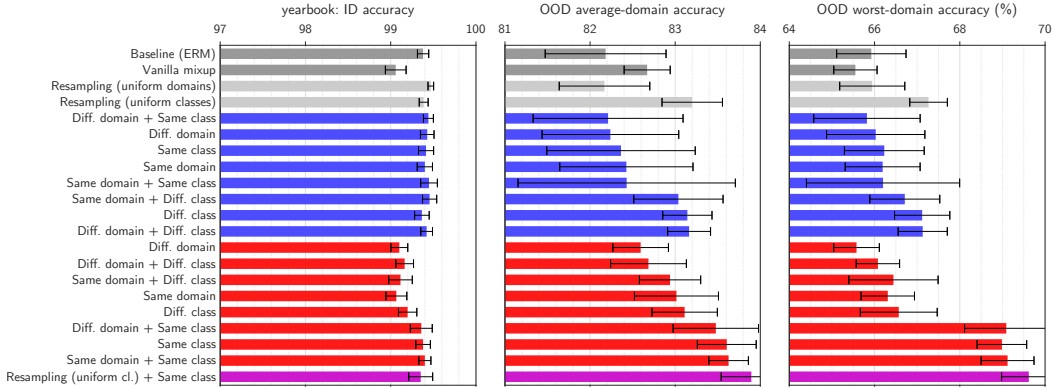

Figure 11: Results on *waterbirds* (top) with linear classifiers on frozen ResNet-50 features and (bottom) with fine-tuned ResNet-50 models (selected methods only).

Figure 12: Results on *yearbook*.

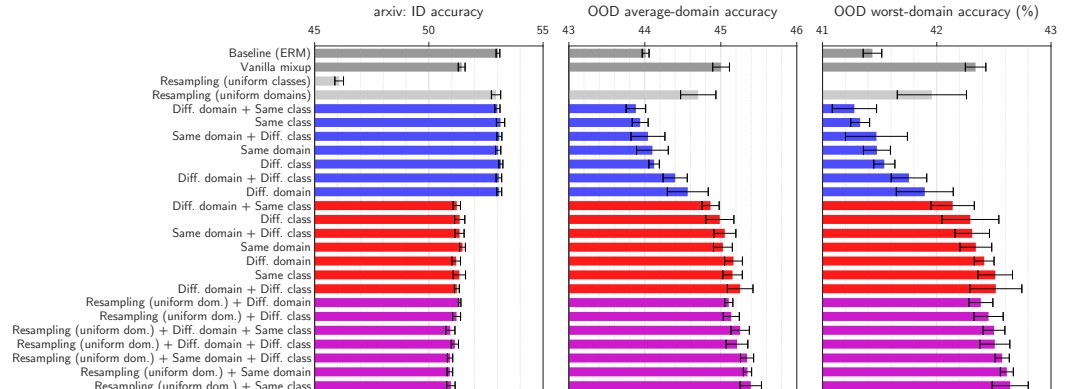

Figure 13: Results on *arXiv*. Interestingly, the methods with selective sampling without mixup are much better than selective mixup on in domain (ID) but worse out of domain (OOD). This shows a clear trade-off between these two objectives.

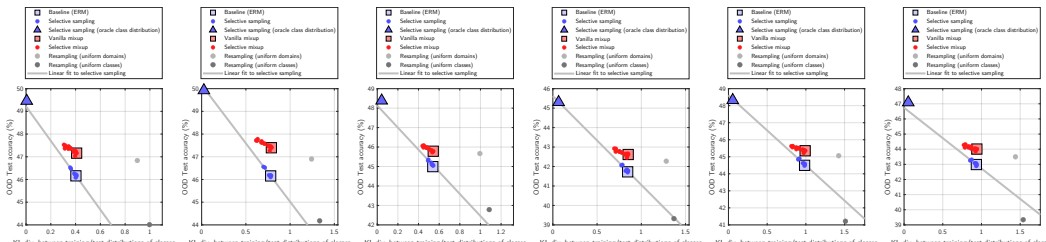

Figure 14: Same analysis as in Figure 7 of the main paper, performed on every test domain. In all cases, we observe a strong correlation between the improvements in accuracy and the reduction in divergence of the class distribution due to resampling effects.

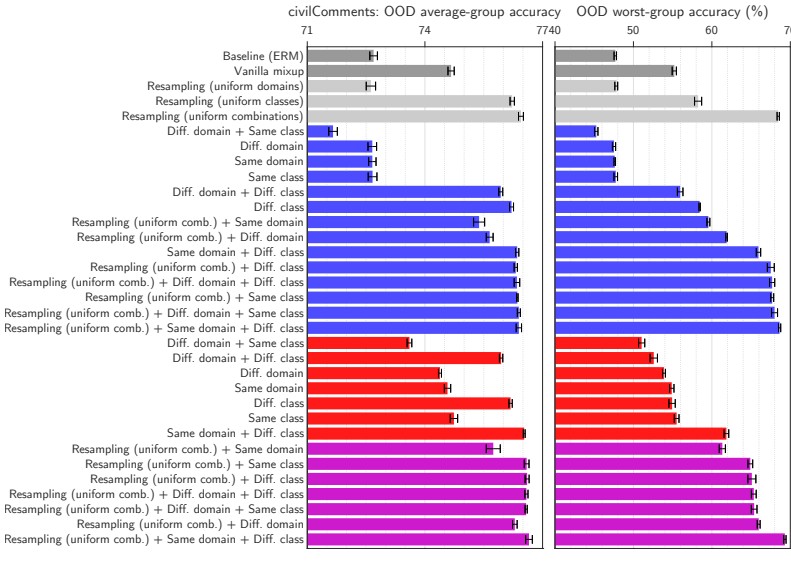

Figure 15: Results on *civilComments* with linear classifiers on frozen embeddings.

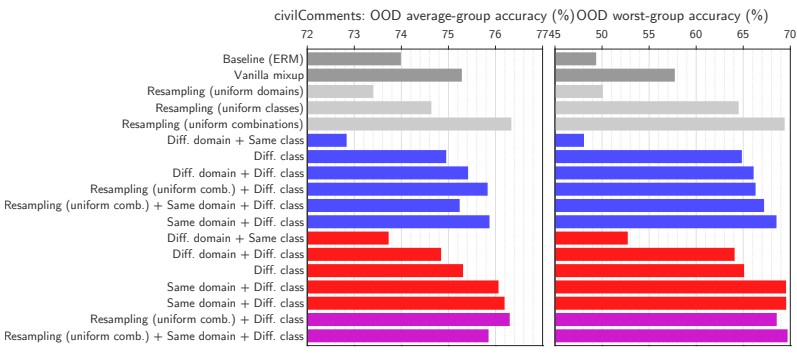

Figure 16: Results on *civilComments* with fine-tuned BERT models (with a single seed). These results are qualitatively identical to those with frozen embeddings above.

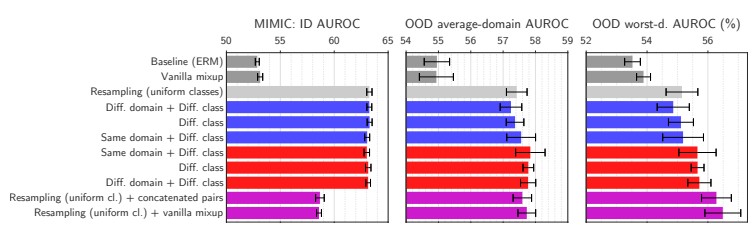

Figure 17: Results on *MIMIC-Readmission*.

## C    Proof of Theorem 3.1

**Theorem C.1** (Restating Theorem 3.1). *Given a dataset $\mathcal{D} = \{(\boldsymbol{x}_i, \boldsymbol{y}_i)\}_i$ and paired data $\widetilde{\mathcal{D}}$ sampled according to the "different class" criterion, i.e. $\widetilde{\mathcal{D}} = \{(\widetilde{\boldsymbol{x}}_i, \widetilde{\boldsymbol{y}}_i) \sim \mathcal{D} \text{ s.t. } \widetilde{\boldsymbol{y}}_i \neq \boldsymbol{y}_i\}$, then the distribution of classes in $\mathcal{D} \cup \widetilde{\mathcal{D}}$ is more uniform than in $\mathcal{D}$.*

*Formally, the entropy $\mathbb{H}\big(\mathbf{p}_Y(\mathcal{D})\big) \leq \mathbb{H}\big(\mathbf{p}_Y(\mathcal{D} \cup \widetilde{\mathcal{D}})\big)$.*

*Proof.* Let us define the shorthands $\boldsymbol{p} \overset{\text{def}}{=} \mathbf{p}_Y(\mathcal{D})$ and $\widetilde{\boldsymbol{p}} \overset{\text{def}}{=} \mathbf{p}_Y(\widetilde{\mathcal{D}})$.

In $\widetilde{\mathcal{D}}$, the $i$th class gets assigned, in the expectation, on a proportion of points equal to the proportion of all other classes $j \neq i$ in the original data $\mathcal{D}$.

Looking at the individual elements of $\widetilde{\boldsymbol{p}}$, we therefore have, $\forall i = 1 \ldots C$:

$$\widetilde{p}_i = \Sigma_{i \neq j}^{C} p_j \qquad\qquad / (C-1) \qquad\qquad (6)$$
$$\widetilde{p}_i = (1 - p_i) \qquad\qquad / (C-1) \qquad\qquad (7)$$

We will show that every element of $\widetilde{\boldsymbol{p}}$ is closer to $\frac{1}{C}$ than the corresponding element of $\boldsymbol{p}$:

$$|p_i - \tfrac{1}{C}| \geq |\widetilde{p}_i - \tfrac{1}{C}| \qquad\qquad (8)$$
$$|\tfrac{C\,p_i - 1}{C}| \geq |\tfrac{(1-p_i)\,C - (C-1)}{C\,(C-1)}| \qquad\qquad (9)$$
$$|C\,p_i - 1| \geq |\tfrac{1 - C\,p_i}{(C-1)}| \qquad\qquad (10)$$
$$|C\,p_i - 1| \geq |\tfrac{C\,p_i - 1}{(C-1)}| \qquad\qquad (11)$$

Therefore $\widetilde{\boldsymbol{p}}$ is closer to a uniform distribution than $\boldsymbol{p}$, and

$$\mathbb{H}(\boldsymbol{p}) \leq \mathbb{H}(\widetilde{\boldsymbol{p}}) \qquad\qquad (12)$$

Since $\mathbf{p}_Y(\mathcal{D} \cup \widetilde{\mathcal{D}}) = \big(\mathbf{p}_Y(\mathcal{D}) \oplus \mathbf{p}_Y(\widetilde{\mathcal{D}})\big) / 2$, we also have

$$\mathbb{H}(\boldsymbol{p}) \leq \mathbb{H}\big((\boldsymbol{p} \oplus \widetilde{\boldsymbol{p}})/2\big) \qquad\qquad (13)$$
$$\mathbb{H}\big(\mathbf{p}_Y(\mathcal{D})\big) \leq \mathbb{H}\big(\mathbf{p}_Y(\mathcal{D} \cup \widetilde{\mathcal{D}})\big) \qquad\qquad (14)$$

with an equality iff $\mathbf{p}_Y(\mathcal{D})$ is uniform. $\qquad\qquad\square$