# OpenReview forum: "Selective Mixup Helps with Distribution Shifts, But Not (Only) because of Mixup"
_NeurIPS.cc/2023/Conference — Submitted to NeurIPS 2023_

### Official Review · Reviewer_SGQh · 2023-06-09

**Soundness:** 3 good
**Presentation:** 4 excellent
**Contribution:** 3 good
**Rating:** 4
**Confidence:** 3

**Summary:**

The authors demonstrate that selective mixup has a similar effect with resampling. This reduces training dataset bias, which may contribute to settings with distribution shifts. Extensive experiments on five datasets were conducted.

**Strengths:**

- A new perspective of (selective) mixup has been discussed.
- Extensive experiments were conducted on various benchmarks
- The paper was well written and was easy to follow.

**Weaknesses:**

The authors have done a great job in providing an extensive list of experiments. However, I have two major concerns:
1. The paper's contribution to the body of knowledge is somewhat weak. I believe the gist of the paper is that "(selective) mixup has a resampling effect". Although this is a novel and interesting interpretation of the mixup data augmentation method, I feel this is not enough compared to the recent standard level of NeurIPS papers. I believe that a more concrete methodology that reflects the authors' findings is needed to qualify as "acceptance level".
2. The experimental results and their discussions are confusing. While the discussion on the waterbirds dataset says that "selective mixup performance is entirely due to resampling", the results on the arxiv dataset says "selective mixup is affected both by vanilla mixup and resampling", and results on the civilComments dataset implies that "resampling is better than mixup". These different experimental results and interpretations do not conclude the effect of resampling and mixup. Rather, the results seem to imply that "mixup and resampling are independent methods that may or may not have correlating effects, depending on what dataset is used". If this is the case, the primary assumption of this paper is undermined.

**Questions:**

In section 4.1, the authors have mentioned that the similar performance indicates that "performance of selective mixup is entirely due to resampling". However, having the same performance level does not mean that one is the cause of the other. Can the authors provide experiments that can demonstrate the causality of the two?

**Limitations:**

Yes

---

> ### Author Rebuttal · Authors · 2023-08-06
>
> > Q1. The paper's contribution to the body of knowledge is somewhat weak. (...) I believe that a more concrete methodology that reflects the authors' findings is needed to qualify as "acceptance level".
>
> We respectfully disagree. This view misses the purpose of science which is to come up with explanations rather than being purely utilitarian [*]. This is key to future progress.
> - This paper overhauls our understanding of a popular method.
> - It connects two hitherto-distinct methods (selective mixup/resampling).
> - This enables better results by better combining the benefits of each.
>
> The key novel finding may seem obvious **in retrospect**, but was completely missed in prior work, despite the widespread use of the method, and multiple years & papers on the topic.
>
> [*] Many influential NeurIPS papers focus on explaining existing methods rather than introducing new algorithms. Examples: [1] *A Universal Law of Robustness via Isoperimetry* (outstanding paper 2021; one reviewer also noted then that "*the tasks and methods are not new*"), [2] *Uniform convergence may be unable to explain generalization in deep learning* (outstanding paper 2019), [3] *An empirical analysis of compute-optimal large language model training* (outstanding paper 2022), [4] *Improved guarantees and a multiple-descent curve for Column Subset Selection and the Nystrom method* (best paper 2020), etc.
>
> A criterion for the NeurIPS awards is specifically: "*Insight — Provides new (and hopefully deep) understanding; does not just show a few percentage points of improvement*".
>
> ----------------------
>
> > Q2. The experimental results and their discussions are confusing. (...) the results seem to imply that "mixup and resampling are independent methods that may or may not have correlating effects, depending on what dataset is used". If this is the case, the primary assumption of this paper is undermined.
>
> There seems to be logical gap in the reviewer's comment. Our factual claims: (in short)
> - selective mixup induces a resampling (the key novel finding),
> - resampling helps with distribution shifts (as known from prior work).
> - mixup helps on some datasets (as known from prior work).
>
> These three points are perfectly supported by empirical evidence: all datasets benefit from the resampling (which was completely missed in prior work on selective mixup) and some datasets **additionally** benefit from (vanilla) mixup. We updated Sec.6 to clarify these takeaways.
>
> ----------------------
>
> > Q3. The authors have mentioned that the similar performance indicates that "performance of selective mixup is entirely due to resampling". (...) Can the authors provide experiments that can demonstrate the causality of the two?
>
> Yes, causality (resampling -> better OOD performance) is supported by:
> 1. The interventional experiments where we independently manipulate the two possible causes (mixing/resampling, cf. bar charts for every dataset).
> 2. The conceptual understanding of the mechanism by which resampling is beneficial under distributon shifts (Sec.3) and the verification that the necessary assumptions hold in the datasets (regression to the mean, Fig.10).
>
> We updated Sec.5 to highlight that all experiments are designed to establish the causal link between resampling and improvements in OOD performance (which prior work could not because of missing important ablations).

---

> > ### Comment · Reviewer_SGQh · 2023-08-16
> > **Rebuttal Acknowledgement**
> >
> > Thank you for your responses. I do agree that a well-written paper does not have to be always utilitarian, and that the authors have done a great job at providing extensive experimental results. However, after reading the paper again, I still have a feeling that the findings are not that significant, and that the conclusions are somewhat hand-wavy. I think it would have been better if the authors could draw a more general conclusion on how selective mixup should be utilized in the perspective of resampling, AND/OR provide further analyses on when/why the author's observations occur on a specific dataset. Simply providing a list of experimental results on different datasets makes it seem more like a technical report than an academic paper to me. For now, I am keeping my score but lowering my confidence.

---

> > > ### Author Response · Authors · 2023-08-16
> > >
> > > Thanks for the discussion. The current presentation might be misleading by overemphasizing the empirical investigations. We propose to revise Sec.1/2 to make it clear that this work is *not empirically driven*. To clarify:
> > >
> > > - This work starts with the conceptual realization that selective mixup induces a non-uniform sampling of the training data (Sec.3, L87-107).
> > > - We then examine theoretically its effect and predict the conditions under which these effects would be observed (regression to the mean, Sec.3 L108-132).
> > > - Only then do we conduct experiments to empirically verify these predictions (Sec.4). While extensive, these experiments are **not** the sole support for our arguments.
> > >
> > > We propose to clarify these points upfront (Sec.1). This will put the empirical investigations in better context and strengthen the message of the paper as more fundamental (i.e. not merely empirical case studies). We also propose to highlight the answer to "how selective mixup should be utilized" (currently in L123-132 **When does one benefit from the resampling**).
> > >
> > > Scientifically, the importance of this work for the community is to provide corrections to erroneous explanations proposed in earlier work, which a whole subfield of research is currently building on.

---

### Official Review · Reviewer_h74Y · 2023-07-04

**Soundness:** 2 fair
**Presentation:** 3 good
**Contribution:** 2 fair
**Rating:** 5
**Confidence:** 3

**Summary:**

The authors focus on explaining the working mechanism of selective Mixup in distribution-shift scenarios. They points out an interesting and novel perspective that selective Mixup benefits not (only) from Mixup itself, but rather the resampling effect it induces. They suggest that such a resampling effect can balance the class and/or domain distribution (depends on the pairing criterion) into uniform distribution, which in turn improves the model's performance.

Theoretically, they show that under the different-class criterion of pairing the data, the classes' distribution does become more uniform. Experimentally, they take the worst-group (class or domain or their combination) performance  as the metric, and conduct experiments on several datasets and models. They investigates a variety of algorithms (e.g. ERM, Mixup, conventional resampling, etc.) and pairing criterions (e.g. different class, different domain, etc.). The results confirms their raised point of view on selective Mixup.

**Strengths:**

1. This paper presents an interesting and novel perspective of  undderstanding the mechanism of selective Mixup. That is, instead of focusing on the data interpolation part of Mixup, which is conventionally considered the key of Mixup and any of its variant, this paper indicates that it's the resampling effect, which averages the data distribution, that enables selective Mixup to success.

2. In all the experiments, the paper has fully investigated all the possible sampling criterions. Furthermore, on the basis of these experiments results, this paper combines the algorithms and criterions that give the best performance respectively and obtains a even better result under its own testing metric.

3. All the figures are well displayed and clearly readable.

**Weaknesses:**

1. This paper is a experiment-driven work. Not many theories are provided to back the main idea up. While Theorem 3.1 shows how certain selective sampling "uniformizes" the class distribution, there is no evidence nor explanation of how it affects the covariate distribution. Also, there is no investigation or explanation of what roles (if any) the Mixup operation itself play in selective Mixup.

2. When performing Mixup and selective Mixup training as baselines, the hyperparameter of the Beta distribution $B(\alpha, \alpha)$ is not carefully determined, or at least the choosing process is not fully explained. If the Mixup baselines failed to reach their optimal states, then the comparison of them with other methods like resampling and selective sampling w/o Mixup might be unqualified.

3. The testing metric only considers the worst-group performance, but not including the overall performance. Normally practitioners are concerned not only with the short board of the bucket, but also all the boards as an entirety (a trade-off between fairness and overall generalization).



**Questions:**

1. Line 77-78, Equation (3). Here the paper suggests that the interpolation coefficient be drawn from $Beta(2,2)$ distribution. Is that also the setting for the following experiments? Since the conventional choice of the $\alpha$ is 1, and some works have indicated that large $\alpha$ may hurt the performance of Mixup. For example, in Figure 2 and line 175-176, the paper implies that selective sampling works similarly whether w/ or w/o Mixup, and that selective Mixup's success is $\textbf{entirely}$ due to resampling; in Figure 8, it also implies that selective sampling works better than selective Mixup. If the $\alpha$ in these experiments are not carefully determined, then these comparison results between selective sampling and selective Mixup may be simply due to the sub-optimality of the latter one baseline. I would suggest that whenever we perform Mixup training as baseline, we select the best hyperparameters setting ($Beta(\alpha,\alpha)$) such that the performance of Mixup is fully recovered in the first place, otherwise the comparison of Mixup with other methods might be unfair to it.

2. Also the impact of manifold intrusion (if exists in the datasets used in this paper) on the Mixup training would be another interesting factor to consider, since this issue itself is a poison that hurts Mixup's performance. Some other works also have implied that Mixup induces label noises into the data. That is to say, if manifold intrusion does exist in some of the datasets, then what selective sampling actually does is simply erasing these poison by removing Mixup operation itself. In this case, it is unreasonable to suggest that resampling operation (and its effect in averaging data distribution) is the key for the success of selective Mixup. On ther other hand, if manifold intrusion does exist, a large $\alpha$ may further exacerbate its impact, which could also explain why sometimes w/o Mixup ourtperforms Mixup significantly.

3. The end of line 187. A redundant right bracket.

4. Figure 3, the second plot (uniform domains) and the fifth (diff. domain any class). The percentages on the figures don't add up to 100%.

5. Figure 2. Is there any insight to explain why vanilla Mixup is the $\textbf{second}$ worst (worse than selective Mixup with diff.domain diff. class criterion)? I was imagining that compared to the latter one, the former one will make the data distribution more imbalanced so it should nevertheless give the worst results.

6. Figure 4. Is there any insight to explain the benefit of same class pairing in Mixup more detailed?

7. CIFAR-10 and CIFAR-100 has some covariate-shift variants, like CIFAR-10-C, CIFAR-100-C, CIFAR-10.1 and CIFAR-10.2. If time and computing resources allow, it would be interesting to see how selective Mixup and selective sampling perform on them.

8. Appendix C, Theorem C.1, Equation (6). The subscript of the summation $\sum_{i\neq{j}}^{C}p_j$ could be rewritten as $j\neq{i}$, which may looks better. Also in both Equations (6) and (7), the typography of the denominator "$/ (C-1)$" is a bit off.

9. Line 120-121. Any insight of how certain selective pairing criterions will also have averaging effect on the covariates ($x$)?


**Limitations:**

1. The paper is fundamentally based on experimental investigation, but with little theoretical support.

2. The datasets used doesn't include some of the most popular or commonly used ones like CIFAR or Imagenet.

3. In the experiments, Mixup (and selective Mixup) are taken as baselines, but the value of $\alpha$ (which supervises the Beta distribution) may not have been carefully determined.

4. I personally imagine that the datasets size and\or the batch size will also have an impact on the performance of selective Mixup and selective sampling. There might be some room for further investigations in the future.

5. What roles (either positive or even negative) the Mixup operation itself plays in selective Mixup exactly have yet to be fully understood.

6. The metric is limited at the worst-group performance with no information on how the models perform in general on all the testing data.

---

> ### Author Rebuttal · Authors · 2023-08-06
>
>
> We believe that all limitations stated by the reviewer are factually incorrect.
>
> > Limitation 1. The paper is fundamentally based on experimental investigation, but with little theoretical support.
>
> This work is not empirically driven. This piece of research starts by conceptually pointing out that selective mixup induces a resampling. We then examine theoretically its effects, making predictions about benefits under distribution shifts and the conditions necessary for these effects. We then perform an extensive set of experiments on standard datasets and verify that these predictions indeed hold precisely when the predicted conditions are met (cf. regression to the mean).
>
>
> > Limitation 2. The datasets used doesn't include some of the most popular or commonly used ones like CIFAR or Imagenet.
>
> We use five standard datasets for evaluation under distribution shifts. ImageNet/CIFAR are unsuitable for this. Variants such as CIFAR-C and CIFAR-10.1 are not widely used in the literature on selective mixup.
>
>
>
> > Limitation 3. In the experiments, Mixup (and selective Mixup) are taken as baselines, but the value of alpha may not have been carefully determined.
>
> This is a good point: we updated the experimental details (Sec.4) to clarify. We did indeed performed a hyperparameter search on the distribution of $\alpha$ for every experiment. Similar to prior work (e.g. [32]) we found very little variation across choices including the standard $\mathcal{B}(2,2)$ or even a fixed $\alpha=0.5$.
>
>
> > Limitation 4. What roles (either positive or even negative) the Mixup operation itself plays in selective Mixup exactly have yet to be fully understood.
>
> The role of the mixing operation (whose theoretical understanding is still poor) is thoroughly examined empirically (Sec.4). We find its effect to vary across datasets (as known from prior work) and to be largely independent from the effects of resampling.
>
> > Limitation 5. The metric is limited at the worst-group performance with no information on how the models perform in general on all the testing data.
>
> The average accuracy is reported for all experiments in Appendix B. The ranking of methods is essentially identical to worst-group performance. We updated Sec.4 to mention it.
>
>
> ----------------------
>
> Thanks for spotting the minor formatting issues.
>
> > Figure 3. Percentages don't add up to 100%.
>
> We checked and these are simply rounding errors.
>
> Minor questions:
>
> > Q1. Figure 2. Is there any insight to explain why vanilla Mixup is the
>  worst (worse than selective Mixup with diff.domain diff. class criterion)?
>
> This seems to align with prior work that showed that mixup is sometimes detrimental to both ID and OOD performance (see related work "*Explaining the benefits of mixup*").
>
> > Q2. Figure 4. Is there any insight to explain the benefit of same class pairing in Mixup more detailed?
>
> This is one case where the original explainations of the LISA method apply [33, Yao et al.]. We can state this more confidently than the original paper thanks to our  ablations that allow separating the effects of the mixing from those of the resampling.
>
> > Q3. L120 Any insight of how certain selective pairing criterions will also have averaging effect on the covariates?
>
> The same reasoning as with labels applies (regression towards "uniformity"). However, it is not clear (to us) what "uniformity" represents in input or feature space. This is an interesting line of inquiry for future work.

---

> > ### Comment · Reviewer_h74Y · 2023-08-21
> > **Rebuttal Acknowledgement**
> >
> > Thank you for the detailed response.

---

### Official Review · Reviewer_hGwy · 2023-07-05

**Soundness:** 2 fair
**Presentation:** 1 poor
**Contribution:** 3 good
**Rating:** 6
**Confidence:** 4

**Summary:**

The paper performs an analysis of selective mixup techniques, coming to the conclusion that some of the improvements these techniques provide can be obtained by the sampling procedure and not the mixing strategy. Based on some of the insights, variants of selective mixup are sometimes proposed to fit specific settings and shown to work better.

**Strengths:**

- The considered datasets and settings are varied and extensive
- The paper proposes plausible explanations of the improved performance of selective mixup variants, and performs extensive ablations and creates experimental settings to test these explanations
- The observations about the roles of re-sampling and regression to the mean are extremely interesting.
- One of the main interesting points of the paper is carefully studying the forms of shift occurring between the training and the test distributions, and studying how specific forms of selective mixup impact them. This methodology can improve the practitioner's understanding of these techniques and how to improve them (e.g. as shown in some of the cases considered).


**Weaknesses:**

- As the authors acknowledge, their analysis is limited to the original mixing strategy.
- The formatting of the paper is a bit bizarre, with plenty of extremely short paragraphs, misplaced image captions etc. I would recommend the authors to fix this issue, as it makes their work look unprofessional.

**Questions:**

- Although the authors frame this as out-of-scope for their work, at least in the cases in which re-sampling and regression to mean are not sufficient to explain the improved performance, it would be very interesting to see how some of the other mixing strategies (using the citations numbers in the paper, [7, 15, 19, 22, 23, 28, 30, 31, 34]) influence the performance. This would give us a preliminary insight on whether perturbing the mixing procedure (e.g. with CutMix, Manifold Mixup, RegMixup etc.) may significantly affect the performance, and if not, future work should identify other factors.
- For Fig. 5 are the authors using Euclidean distances? If so, the result may not be particularly meaningful. Could the authors check the same pattern that occurs with other distance measures in the input space? (e.g. using [1,2])


[1] https://github.com/richzhang/PerceptualSimilarity
[2] https://github.com/Taited/clip-score

**Limitations:**

Adequately Addressed

---

> ### Author Rebuttal · Authors · 2023-08-06
>
> > Q1. analysis limited to the original mixing strategy
>
> Most alternative mixing strategies are image-specific (e.g. CutMix) and unsuitable for other domains (CivilComments, arXiv, MIMIC, etc.). For yearbook, we follow the standard implementation of prior work [32]: the low-resolution images are unsuitable for methods like CutMix. For waterbirds, Appendix B compares the main methods on standard mixup (in input space) and manifold-mixup (on features from a frozen ResNet): results are essentially identical.
>
>
> > Q2. For Fig. 5 are the authors using Euclidean distances? If so, the result may not be particularly meaningful. Could the authors check the same pattern that occurs with other distance measures in the input space? (e.g. using [1,2])
>
> We have indeed performed the same experiment with distances over learned features (similar to perceptual similarity [1]) and obtain essentially identical results. The reviewer is correct that this is not obvious a priori. This holds for the yearbook dataset because all images represent aligned faces in grayscale and low resolution. We clarified this in Sec.4.2 and Appendix A.
>
>
> > formatting of the paper is a bit bizarre
>
> The paper follows the NeurIPS LaTeX style. We admit to a different writing style from many papers but this *cannot be seen as a critical weakness*.

---

> > ### Comment · Reviewer_hGwy · 2023-08-10
> > **Thank you for your rebuttal**
> >
> > I have read the authors rebuttals and other reviews.
> >
> > **Regarding other review's weakness on not proposing a methodology or with an extensively developed theory**: I respectfully disagree with other reviewers, as I think this is clearly a paper developing insights/analyses and not proposing a methodology. These papers, if the analysis is insightful and interesting, are as worthy of acceptance as well as papers proposing methodologies with (allegedly) improved performance. Therefore I do not consider the lack of novel methodology a weakness at all, and some other reviewers agree it is an interesting take on Mixup techniques.
> >
> > Furthermore, many theoretical works on Mixup propose theories that have been dropped in following research, as they have not been found to be particularly grounded or useful. A good empirical analysis is valuable independently of the presence of lengthy theoretical discussion, often based on simplified and unrealistic assumptions or conjecturing the existence of handwavy notions that spread through the literature despite being vague (e.g. the mentioned manifold intrusion). Therefore I do not change my opinion about the value of the work proposed.
> >
> > **Regarding the rebuttal to my review**:
> > - On different mixing strategies: Even restricting the domain to higher resolution images for which these mixing strategies make sense, it would be an interesting exploratory analysis.
> > - On similarities: If feature embeddings similarities of standard extractors were sufficient, LPIPS and CLIP scores would not be used in the literature of generative models. While those two metrics do not constitute a solid ground-truth for perceptual similarities, I still believe experiments showing them would be more convincing.
> > - About formatting: it's obviously not a critical weakness, but it would be better to improve it, especially because most of the broken paragraphs are not conducive of a better graphical organisation of your work. Unusual layout makes sense if it makes the presentation more effective. I do not believe it does. This point has very little relevance to my final score.
> >
> > For the moment, I maintain my score.

---

### Official Review · Reviewer_xtzm · 2023-07-07

**Soundness:** 3 good
**Presentation:** 3 good
**Contribution:** 2 fair
**Rating:** 4
**Confidence:** 4

**Summary:**

This paper presented an insight that selective mixup (different class or different domain) is actually equivalent to resampling. A simple mathematical proof is presented showing that the mixed up distribution is closer to uniform distribution in terms of label distribution. Extensive experimental results are presented to demonstrate the findings.

**Strengths:**

* novel understanding on selective mixup
* extensive experiments to demonstrate the understanding


**Weaknesses:**

The finding is nice but seems value is limited, at least for two-class problems, selective mixup obviously leads to uniform distribution. This paper is more about empirical demonstration.

**Questions:**

Can I summarize the finding as: selective mixup is mostly equivalent to resampling, and the mixing operation only helps if vanilla mixup helps?

minor question: for waterbird dataset, if class and domain are reversely correlated, how do you sample different classes from the same domain? or is it not perfect correlation?


**Limitations:**

not new algorithm, but understanding existing algorithm, the value of the finding seems limited.

---

> ### Author Rebuttal · Authors · 2023-08-06
>
> > Q1. not new algorithm, but understanding existing algorithm, the value of the finding seems limited
>
> We respectfully disagree with this comment. The point of science is not only proposing new algorithms but **understanding** how things work. This paper overhauls our understanding of a popular method (selective mixup) which is one of the most successful to address distribution shifts (cf. WILDS & Wild-Time leaderboards). Understanding its mechanisms is critical for further progress in this field.
>
>
> > Q2. Can I summarize the finding as: selective mixup is mostly equivalent to resampling, and the mixing operation only helps if vanilla mixup helps?
>
> Not exactly, this summarizes the empirical observations but not the methods (which, again, is what matters to scientific understanding). Selective mixup is not "*mostly equivalent*" to resampling. More correctly: selective mixup induces an implicit resampling that explains some of its effects.
>
>
> > Q3. minor question: for waterbird dataset, how do you sample different classes from the same domain? or is it not perfect correlation?
>
> Indeed, not perfect correlation (as shown in Fig.3 leftmost panel).
>
>
> We invite the reviewer to point out actual errors, conceptual mistakes, or overlap with existing literature (if any) to justify the proposed rating.

---

> > ### Comment · Reviewer_xtzm · 2023-08-17
> >
> > I highly appreciate understanding work, and the findings are nice, but I'm still not convinced they are significant and insightful enough for acceptance, but I would not be upset if it's accepted, I'll leave to AC for decision. Thanks!

---

> > > ### Author Response · Authors · 2023-08-18
> > >
> > > Thanks. Regarding the significance of the findings, let us highlight 3 points.
> > >
> > > 1. The literature on Mixup is extensive and relevant to a large community: the original paper has been cited >7000 times and spawned a entire subfield of research [Zhang et al., ICLR'18].
> > > 2. The focus on OOD generalization is particularly important: selective mixup is one of the most successful approaches (see the DomainBed, WILDS, and WILD-Time benchmarks) among a plethora of competing methods, and distribution shifts being arguably one of the main challenges in ML today.
> > > 3. Despite this wide interest, the understanding of Mixup is far from complete. **We lift some of the mistery on the effectiveness of selective Mixup for OOD generalization**. We discovered that much of the improvements reported in prior work are mediated by a side "resampling" effect. This mode of action was completely overlooked in prior work, and we reveal that **existing studies lacked the proper ablations** to account for these effects, leading to erroneous explanations and conclusions. Practically, we show that much of the improvements can be achieved with much simpler resampling/reweighting, a well-known methods to address data imbalance. With these findings, we connect two subfields of research thought so far to be completely disconnected. And our arguments are based on both a theoretical analysis and an extensive empirical validation.
> > >
> > > A whole subfield of research is currently building on selective mixup (see our literature review). Therefore, **corrections to erroneous explanations/methodologies in the existing literature are critical for further progress**.
> > >
> > > We are open to suggestions to better highlight these points in the manuscript if needed. We hope this helps comitting to a firm reviewing decision. Thanks for your time!

---

### Author Response · Authors · 2023-08-20
**Looking forward to more discussions**

Dear all reviewers,

As the deadline for author-reviewer discussions is approaching, we would like to discuss more with you on our rebuttal. If you have any further questions, please let us know.

Thanks

---

### Decision · Program_Chairs · 2023-09-21

**Decision:**

Reject

**Comment:**

There are two borderline rejects, one weak accept and one borderline accept. AC read through the paper and find it somewhat interesting but this is not a actionable paper but an insight paper. The basic finding is that selective mixup is "nothing more than regression towards the mean", and experiments were provided to show that. Due to the disparate ratings and questions still remain on its utility, AC recommends that this paper be improved and submitted to the next conference.